# A mechanically adaptive hydrogel with a reconfigurable network consisting entirely of inorganic nanosheets and water

Koki Sano [1✉], Naoki Igarashi[1,2], Yasuo Ebina [3], Takayoshi Sasaki [3], Takaaki Hikima[4], Takuzo Aida [1,2✉] & Yasuhiro Ishida [1✉]

Although various biomimetic soft materials that display structural hierarchies and stimuli responsiveness have been developed from organic materials, the creation of their counterparts consisting entirely of inorganic materials presents an attractive challenge, as the properties of such materials generally differ from those of living organisms. Here, we have developed a hydrogel consisting of inorganic nanosheets (14 wt%) and water (86 wt%) that undergoes thermally induced reversible and abrupt changes in its internal structure and mechanical elasticity (23-fold). At room temperature, the nanosheets in water electrostatically repel one another and self-assemble into a long-periodic lamellar architecture with mutually restricted mobility, forming a physical hydrogel. Upon heating above 55 °C, the electrostatic repulsion is overcome by competing van der Waals attraction, and the nanosheets rearrange into an interconnected 3D network of another hydrogel. By doping the gel with a photothermal-conversion agent, the gel-to-gel transition becomes operable spatiotemporally on photoirradiation.

[1] RIKEN Center for Emergent Matter Science, 2-1 Hirosawa, Wako, Saitama 351-0198, Japan. [2] Department of Chemistry and Biotechnology, School of Engineering, The University of Tokyo, 7-3-1 Hongo, Bunkyo-ku, Tokyo 113-8656, Japan. [3] National Institute for Materials Science, International Center for Materials Nanoarchitectonics, 1-1 Namiki, Tsukuba, Ibaraki 305-0044, Japan. [4] RIKEN SPring-8 Center, 1-1-1 Kouto, Sayo, Hyogo 679-5198, Japan. ✉email: koki.sano@riken.jp; aida@macro.t.u-tokyo.ac.jp; y-ishida@riken.jp

In scientific fiction and mythology, inorganic creatures often appear. These stories present us with a new challenge in materials science of how to produce life-like materials entirely from inorganic substances. This seems considerably difficult because living organisms usually consist of water-rich flexible solids with a structural hierarchy that are capable of responding to stimuli, unlike inorganic materials, which generally exhibit poor processability, low flexibility, and a lack of responsiveness. Indeed, previous studies on biomimetic soft materials have almost exclusively involved the use of organic constituents[1–8], as typically represented by mechanically adaptive hydrogels based on thermoresponsive organic polymers such as poly(*N*-isopropylacrylamide)[7–13]. However, if counterparts of these hydrogels consisting entirely of inorganic materials were to become available, they could expand the scope of materials science by adopting a complementary role to organic-based biomimetic soft materials in providing such characteristics as good mechanical properties, long-term durability, and low environmental burdens[14–16].

For the creation of inorganic-based biomimetic soft materials, the most promising constituents are inorganic nanomaterials such as nanoparticles, nanofibers, or nanosheets[17]. When these inorganic materials self-assemble to form 3D architectures in water, they sometimes afford hydrogels[18–25]. In most cases, the inorganic nanomaterials aggregate through attractive interactions, such as van der Waals attraction or electrostatic attraction, to form interconnected 3D networks that can hold large quantities of water; these materials are referred to as attraction-dominant gels (Fig. 1e–g)[18–22]. However, inorganic nanomaterials can also participate in another type of self-assembly to produce a 3D architecture that is capable of exhibiting a gel-like behavior, although previous examples are limited to only a few examples on charged inorganic nanosheets[23–25]. If charged nanosheets exhibit a sufficiently strong mutual electrostatic repulsion, they can spontaneously self-assemble into a long-periodicity lamellar architecture in which their mobility is restricted. As a result, their aqueous dispersions can exhibit a gel-like behavior and such materials are referred to as repulsion-dominant gels (Fig. 1b–d).

Although these two types of inorganic-based hydrogel have been studied separately by using various nanomaterials, we conjectured that a single aqueous system containing charged nanosheets might be capable of reversibly forming both types of hydrogel through careful tuning of the intensity of the competing attractive and repulsive forces according to the Derjaguin–Landau–Verwey–Overbeek (DLVO) theory[26,27]. Such a hydrogel might be endowed with responsiveness to stimuli, despite its inorganic composition.

In this work, by adopting the above strategy, we succeeded in developing a stimuli-responsive hydrogel consisting of anionic nanosheets of titanate[27–30] (TiNSs; 14 wt%) and water (86 wt%) that, depending on the temperature, can reversibly adopt one of two hydrogel states: a repulsion-dominant state or an attraction-dominant state (Fig. 1). In this gel-to-gel transition driven by tuning of the electrostatic repulsion between TiNSs, the topology of the internal structure of the hydrogel changes abruptly (Fig. 2). Owing to the 2D shape of TiNSs, which prevents the possibility of entanglement, and the particular thermoresponsiveness of TiNSs, the gel-to-gel transition occurs rapidly with little hysteresis and without deterioration upon repetition (Fig. 3). Because of the abrupt topological reconfiguration of the internal structure of the material, the gel-to-gel transition is accompanied by a 23-fold change in the hydrogel's mechanical elasticity, reminiscent of that of sea cucumbers (Fig. 4)[31,32]. Furthermore, by doping with a small number of gold nanoparticles that act as photothermal converters[33], the hydrogel can become responsive to photo stimuli[33–36], permitting the gel-to-gel transition to occur in a

spatiotemporally controlled manner (Fig. 5). Importantly, it is usually difficult to construct a gel that displays such a topological reconfiguration, even by using organic materials[37,38].

## Results

### Preparation and characterization of a hydrogel consisting of titanate nanosheets in the repulsion-dominant state (TiNS-Gel$_{Repuls}$).
The key to our achievement was the use of a particular nanosheet of titanate (TiNS) characterized by its ultra-thin (0.75 nm) and extra-wide (several micrometers) dimensions (Fig. 1a and Supplementary Fig. 1)[27–30]. TiNS is densely populated with negative charges (1.5 C m$^{-2}$) that are surrounded by tetrabutylammonium countercations. By using TiNS as a constituent, a repulsion-dominant hydrogel, denoted as TiNS-Gel$_{Repuls}$, was prepared as follows. A 0.4 wt% aqueous dispersion of TiNS (30 mL) was centrifuged at 20,000 *g* for 1 h, so that the dispersion became segregated into a TiNS-free supernatant (~29 mL) and a water-rich sediment ([TiNS] = 14 wt%; 0.86 mL). In this sediment, TiNSs were confined close to one another so that their mutual electrostatic repulsion became strong enough to restrict their mobility. Consequently, the sediment existed as a physical gel (Fig. 1b–d; TiNS-Gel$_{Repuls}$)[23–25] that could be isolated from the supernatant and removed from the centrifugation tube. Upon dilution with water, TiNS-Gel$_{Repuls}$ retained its gel-like behavior up to a TiNS concentration of 8 wt%, but turned into a sol on further dilution (Supplementary Fig. 2).

TiNS-Gel$_{Repuls}$ thus obtained showed structural profiles consistent with its supposed gelation mechanism as described above (Fig. 2a). In small-angle X-ray scattering (SAXS) measurements at 25 °C, TiNS-Gel$_{Repuls}$ exhibited multiple scattering peaks with *q* value ratios of 1, 1/2, 1/3, …, and 1/*x*, suggesting the formation of a lamellar architecture of TiNSs with a uniform and large interlamellar distance of 11.8 nm (Fig. 2b). This distance is 16 times larger than the thickness of TiNS (0.75 nm), indicating that TiNSs were not in contact due to their mutual electrostatic repulsion (Fig. 1d). In a polarized optical microscopy (POM) study at 25 °C, TiNS-Gel$_{Repuls}$ exhibited a strong birefringence under crossed Nicols (Fig. 2c), which also confirmed the existence of a liquid-crystalline lamellar architecture in TiNS-Gel$_{Repuls}$ (Fig. 1c). Furthermore, scanning electron microscopy (SEM) observations of a freeze-dried sample of TiNS-Gel$_{Repuls}$ showed the presence of a layered arrangement of TiNSs (Fig. 2d), probably originating from a lamellar architecture of TiNSs before freeze drying.

### Preparation and characterization of a hydrogel consisting of titanate nanosheets in the attraction-dominant state (TiNS-Gel$_{Attract}$).
We previously reported that the electrostatic repulsion between TiNSs becomes attenuated upon raising the temperature[27]. Although this tendency had been observed in aqueous TiNS dispersions, we conjectured that a similar phenomenon might occur in a hydrogel of TiNSs (TiNS-Gel). Indeed, when TiNS-Gel$_{Repuls}$ was heated from 25 to 90 °C, it changed into another type of hydrogel with a different internal structure and different physical properties (Fig. 2e–h). Throughout this transition, exudation of water from the hydrogel was not observed at all, suggesting that the volume of the hydrogel remained. Because the formation of this hydrogel was dominated by attractive forces between TiNSs, as later discussed (see Fig. 6a–c below), this hydrogel is denoted as TiNS-Gel$_{Attract}$.

In the SAXS profiles of TiNS-Gel$_{Attract}$ at 90 °C, the sets of diffractions due to the lamellar architecture of TiNSs with a periodicity of 11.8 nm, characteristic of TiNS-Gel$_{Repuls}$, disappeared, while a single diffraction corresponding to a periodicity of 2.6 nm emerged (Fig. 2f). The new periodicity was similar to that

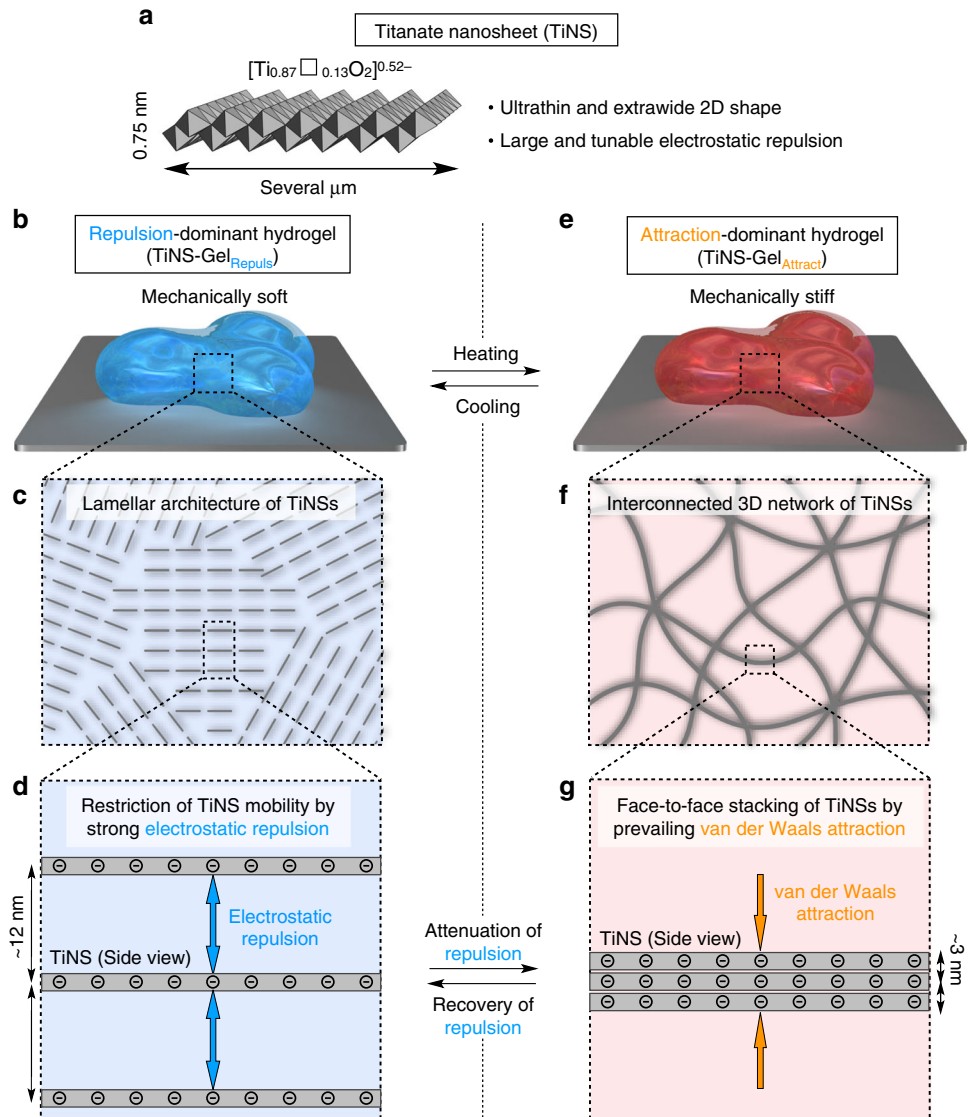

**Fig. 1 Thermoresponsive hydrogel consisting of inorganic nanosheets (titanate nanosheet; TiNS) and water. a** A schematic illustration of unilamellar titanate (IV) nanosheet (TiNS). Countercations are omitted for clarity. Open square indicates vacant sites. **b–g** Schematic illustrations of the hydrogel of TiNS (TiNS-Gel) in a repulsion-dominant state (TiNS-Gel$_{Repuls}$; **b–d**) and an attraction-dominant state (TiNS-Gel$_{Attract}$; **e–g**). When the electrostatic repulsion between TiNSs in an aqueous dispersion is strong enough, TiNSs spontaneously self-assemble into a long-periodicity lamellar architecture (**c**) in which their mobility is mutually restricted (**d**). As a result, their aqueous dispersion can exhibit a gel-like behavior, denoted as TiNS-Gel$_{Repuls}$. When TiNS-Gel$_{Repuls}$ is heated above a critical temperature, the electrostatic repulsion becomes weaker than the competing van der Waals attraction, so that TiNSs abruptly stack tightly (**g**) to form an interconnected 3D network that can hold large quantities of water (**f**), denoted as TiNS-Gel$_{Attract}$. Because of the large difference in the topology of the internal structure between TiNS-Gel$_{Repuls}$ and TiNS-Gel$_{Attract}$, this gel-to-gel transition is accompanied by drastic changes in the optical and mechanical properties.

of aggregates of TiNSs prepared by drying an aqueous TiNS dispersion ([TiNS] = 0.4 wt%) at a relative humidity of 95% for 12 h ($d$ = ~2.7 nm)[28]. By taking account of the SAXS profiles of TiNS-Gel$_{Attract}$ (Fig. 2f), together with the fact that the transition from TiNS-Gel$_{Repuls}$ to TiNS-Gel$_{Attract}$ proceeded without a change in the gel volume, the internal structure of TiNS-Gel$_{Attract}$ is considered to consist of an interconnected 3D network formed by partial stacking of TiNSs (Fig. 1f, g). In POM observations at 90 °C, TiNS-Gel$_{Attract}$ did not show any birefringence (Fig. 2g), indicating that its internal 3D structure is macroscopically isotropic, unlike that of TiNS-Gel$_{Repuls}$. SEM observations of a freeze-dried sample of TiNS-Gel$_{Attract}$ confirmed that the internal 3D structure of TiNS-Gel$_{Attract}$ was markedly different from that of TiNS-Gel$_{Repuls}$ (Fig. 2h).

**Thermoresponsive transition between TiNS-Gel$_{Repuls}$ and TiNS-Gel$_{Attract}$.** We then performed a detailed investigation of how the transition between TiNS-Gel$_{Repuls}$ and TiNS-Gel$_{Attract}$ occurs. First, we examined TiNS-Gel ([TiNS] = 14 wt%) by differential scanning calorimetry at temperatures between 25 and 90 °C at a scanning rate of 1 °C min$^{-1}$ (Fig. 3a). A sharp single peak was observed at ~55 °C during both heating and cooling of the gel (Fig. 3b), confirming that the transition between TiNS-Gel$_{Repuls}$ and TiNS-Gel$_{Attract}$ is a first-order phase transition that occurs at ~55 °C.

To clarify how the internal structure changed in the gel-to-gel transition, we monitored the SAXS profiles of TiNS-Gel ([TiNS] = 14 wt%) while performing a temperature sweep between 25 and 90 °C (Fig. 3c and Supplementary Fig. 3). As described above (Fig. 2b), TiNS-Gel$_{Repuls}$ at 25 °C showed a set of diffractions

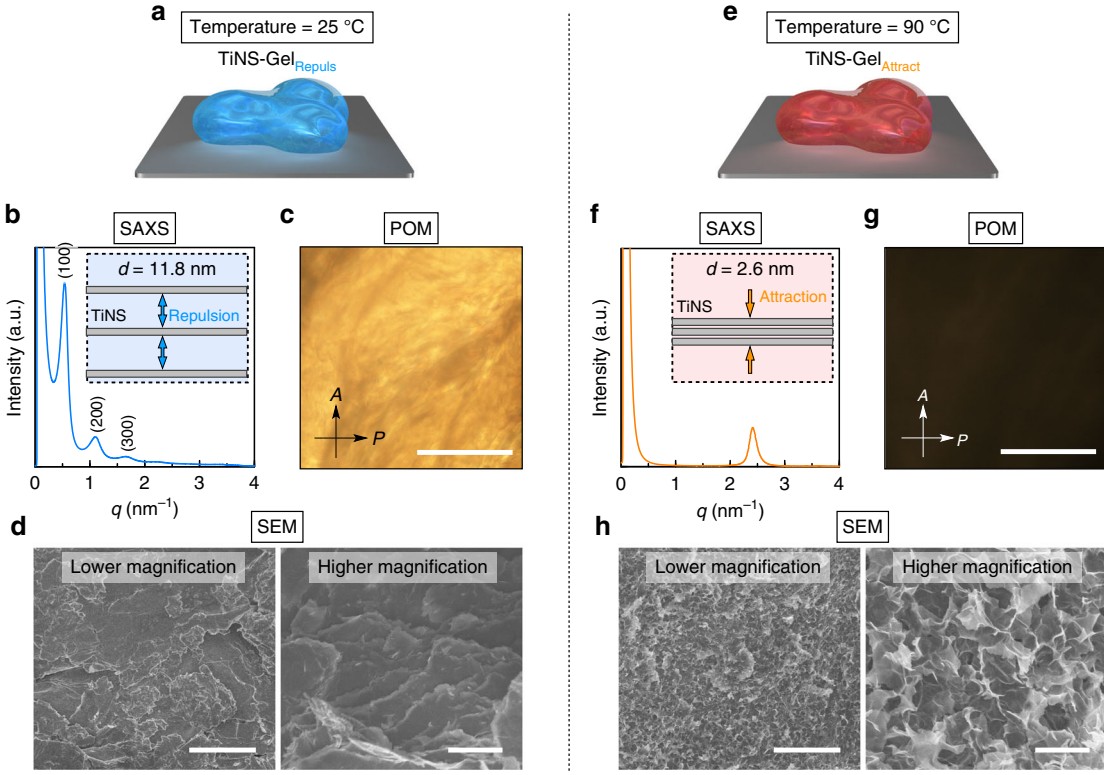

**Fig. 2 Internal structures of TiNS-Gel_{Repuls} and TiNS-Gel_{Attract}. a, e** Schematic illustrations of the hydrogel of TiNS (TiNS-Gel) in a repulsion-dominant state (TiNS-Gel_{Repuls}; **a**) and an attraction-dominant state (TiNS-Gel_{Attract}; **e**). **b, f** Small-angle X-ray scattering (SAXS) profiles of TiNS-Gel ([TiNS] = 14 wt %) at 25 °C (**b**) and 90 °C (**f**). Insets indicate a schematic illustration of the nanosheet arrangements at a nanoscale level. **c, g** Polarized optical microscopy (POM) images under crossed Nicols of TiNS-Gel ([TiNS] = 14 wt%) at 25 °C (**c**) and 90 °C (**g**). **d, h** Scanning electron microscopy (SEM) images of freeze-dried samples of TiNS-Gel ([TiNS] = 14 wt%) prepared by rapid freezing from 25 °C (**d**) and 90 °C (**h**) using liquid nitrogen. Scale bars: 1 mm (**c, g**); 50 μm (**d, h**, left); 10 μm (**d, h**, right).

ascribable to a lamellar architecture of TiNSs with a periodicity of 11.8 nm (Fig. 1d). Upon heating from 25 to 50 °C, this set of diffractions shifted gradually to the wider-angle region due to shrinkage of an interlamellar periodicity to ~8 nm (Fig. 3c). When the temperature reached 55 °C, the set of diffractions suddenly disappeared and a single diffraction corresponding to a periodicity of ~3 nm emerged, indicating transition into TiNS-Gel_{Attract}. Upon further heating to 90 °C, the new diffraction peak shifted slightly to the wider-angle region possibly because of the heat-induced dehydration of countercations of TiNSs and/or TiNS surfaces[39]. In the cooling process from 90 to 25 °C, the reverse changes took place. Even when this heating/cooling cycle was repeated, the temperature-dependent SAXS profiles remained largely unchanged (Supplementary Fig. 3).

To investigate the rate of the gel-to-gel transition, we subjected TiNS-Gel_{Repuls} ([TiNS] = 14 wt%) to an abrupt change in temperature from 25 to 90 °C (Fig. 3d) and we monitored the time-course changes in its birefringence by using POM under crossed Nicols. The initial bright POM image darkened within 2 s (Fig. 3e), confirming the rapidity of the transition from TiNS-Gel_{Repuls} into TiNS-Gel_{Attract} (Fig. 3f).

To summarize, in response to a temperature change, the internal network topology of TiNS-Gel switches quickly and sharply within a narrow temperature range with almost no hysteresis or deterioration upon repetition. Although the gel-to-gel transition involves a marked topological reconfiguration of the internal structure, it proceeds with excellent speed and reversibility, probably because the 2D shape of TiNSs eliminates the possibility of entanglement.

**Mechanical properties of the TiNS hydrogel.** Sea cucumbers, upon external stimulation, can abruptly switch their mechanical properties by modulating their internal structures[31,32]. This phenomenon has attracted significant attention from both fundamental and practical viewpoints. We conjectured that our TiNS-Gel might be capable of mimicking this function, because its gel-to-gel transition proceeded with excellent rapidity and perfect reversibility, and involves drastic topological reconfiguration of the internal structure of the gel.

To confirm this possibility, we conducted rheological tests on TiNS-Gel ([TiNS] = 14 wt%) at 25 and 90 °C, where it exists as TiNS-Gel_{Repuls} and TiNS-Gel_{Attract}, respectively. In a frequency sweep (frequency = 0.1–10 rad s$^{-1}$; strain = 0.1%) at 25 °C, TiNS-Gel_{Repuls} showed storage moduli ($G'$) that were higher than its loss moduli ($G''$) over the entire frequency range, confirming its gel-like properties, where $G'$ at a frequency of 1.0 rad s$^{-1}$ was 26 kPa (Fig. 4a). When the same setup was heated to 90 °C, the resultant TiNS-Gel_{Attract} also showed a similar gel-like property with $G' > G''$ over the entire frequency region, but the $G'$ value was more than 20 times larger than that of TiNS-Gel_{Repuls} ($G' = 600$ kPa at a frequency of 1.0 rad s$^{-1}$; Fig. 4b). When a heating/cooling cycle between 25 and 90 °C was repeated three times, the thermo-responsive changes in the $G'$ and $G''$ values were barely affected (Fig. 4c), demonstrating the excellent reversibility and rapidity of this mechanical-property switching.

Compression tests also confirmed the ability of TiNS-Gel to change its mechanical properties in response to thermal stimuli. The Young's modulus of TiNS-Gel ([TiNS] = 14 wt%) increased from 1.3 to 20 kPa when it underwent a transition from TiNS-

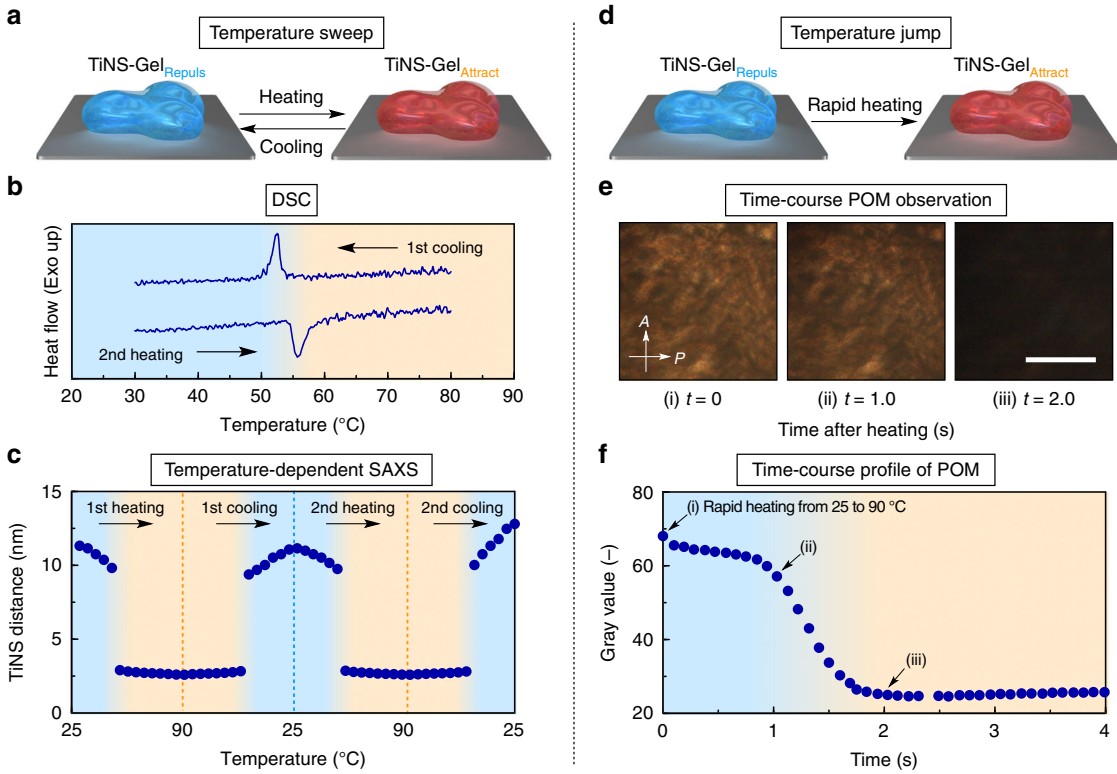

**Fig. 3 Reversible and rapid transition between TiNS-Gel_Repuls and TiNS-Gel_Attract in response to thermal stimuli. a** A schematic illustration of the TiNS-Gel behavior upon temperature sweep. **b** Differential scanning calorimetry (DSC) measurement of TiNS-Gel ([TiNS] = 14 wt%) between 25 and 90 °C at a rate of 1 °C min⁻¹. **c** Changes in the TiNS distance of TiNS-Gel ([TiNS] = 14 wt%) upon repeated thermal scanning between 25 and 90 °C. The TiNS distance was determined by SAXS measurements. **d** A schematic illustration of the TiNS-Gel behavior upon temperature jump. **e, f** Time-course POM images (**e**) and time-course plots of the gray value of the POM images (**f**) of TiNS-Gel ([TiNS] = 14 wt%) after an abrupt temperature jump from 25 to 90 °C. Scale bar: 1 mm (**e**).

Gel_Repuls into TiNS-Gel_Attract on heating from 25 to 90 °C (Supplementary Fig. 4).

**Spatiotemporal controllability of a TiNS hydrogel.** The results reported above prompted us to attempt to produce a more life-like smart material from TiNS-Gel by expanding the range of stimuli capable of inducing the gel-to-gel transition. In particular, the use of light as a stimulus is an attractive option that should permit the gel-to-gel transition to occur in a spatiotemporally controlled manner (Fig. 5a), similar to some biological systems[33–37]. Because TiNS-Gel itself is thermoresponsive, it might be modified to produce a photoresponsive version by the addition of an appropriate photothermal converter.

As a photothermal converter, we chose gold nanoparticles (AuNPs) because of their water dispersibility, compatibility with TiNSs, and excellent photothermal-conversion ability[33]. Thus, TiNS-Gel_Repuls ([TiNS] = 14 wt%) was doped with monodisperse AuNPs ([AuNP] = 0.05 wt%) with an average diameter of 17 nm (Fig. 5b, c). The mechanical properties and thermoresponsiveness of AuNP-doped TiNS-Gel were well preserved (Supplementary Fig. 5). Also, the photothermal-conversion ability of AuNPs was preserved throughout the hybridization; when AuNP-doped TiNS-Gel was irradiated with 445-nm laser light (power density = 5.6 W cm⁻²), the temperature of the irradiated region increased from 25 to 82 °C within 40 s, as monitored by a thermal-imaging camera (Fig. 5d, right). Because AuNP-free TiNS-Gel did not show any temperature change under the same photoirradiation condition (Fig. 5d, left), the present photothermal conversion can be ascribed to the effects of AuNPs exclusively. When the light was turned off, the irradiated region of AuNP-doped TiNS-Gel

was air-cooled to 25 °C within 60 s. The light-on/light-off cycle could be repeated many times without detectable deterioration (Fig. 5e).

As we envisioned in Fig. 5a, POM observation of these temperature-changing processes showed that AuNP-doped TiNS-Gel underwent a transition between TiNS-Gel_Repuls and TiNS-Gel_Attract rapidly, reversibly, and in a spatiotemporally controlled manner (Fig. 5f). On photoirradiation of TiNS-Gel_Repuls with 445-nm laser light (power density = 5.6 W cm⁻²; irradiated region = 2 × 4 mm) for 40 s, the irradiated region was selectively transformed into TiNS-Gel_Attract with no birefringence, whereas upon ceasing photoirradiation, the irradiated region returned to TiNS-Gel_Repuls with a large birefringence within 4 s (Supplementary Movie 1). Detailed studies revealed that (i) the transition from TiNS-Gel_Repuls to TiNS-Gel_Attract under this condition was finished within ~2 s and (ii) the transition speed can be tuned by changing the light power density (Supplementary Fig. 6b) and AuNP concentration (Supplementary Fig. 6c). This dynamic structural change, which is highly controllable in both time and space, is reminiscent of the responsive behaviors of living organisms, despite the inorganic composition of TiNS-Gel.

**Key parameters for TiNSs to form a mechanically adaptive hydrogel.** Although many hydrogels consisting entirely of such inorganic nanomaterials as nanoparticles, nanofibers, or nanosheets have been developed, TiNS-Gel is the only known hydrogel that is capable of reversibly adopting one of two hydrogel states. To clarify which are the key factors underlying this property, we conducted systematic studies on the characteristic profiles of TiNSs in terms of (i) its large and tunable

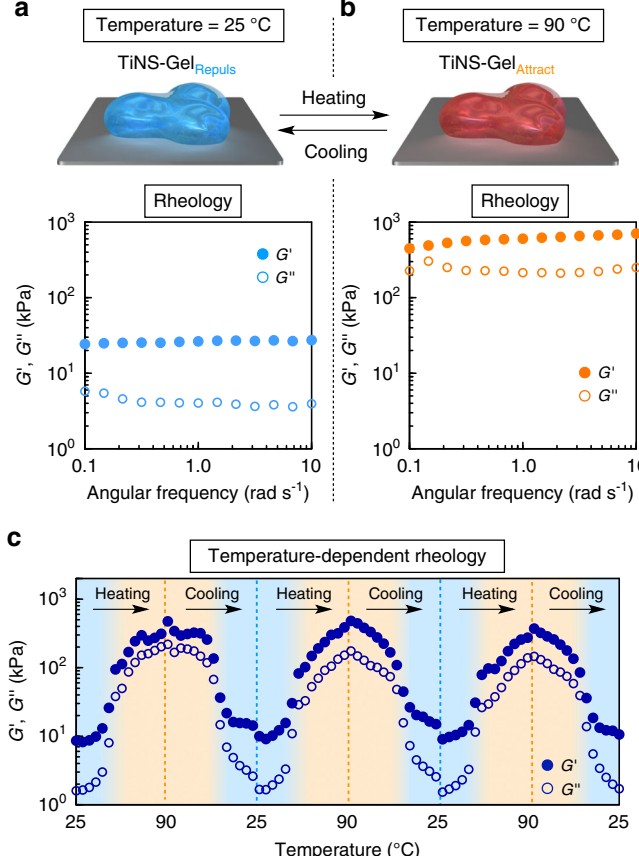

**Fig. 4 Thermoresponsive switching of the mechanical properties of TiNS-Gel through the transition between TiNS-Gel$_{Repuls}$ and TiNS-Gel$_{Attract}$. a, b** Storage ($G'$) and loss ($G''$) moduli on frequency sweep (0.1–10 rad s$^{-1}$) at a fixed strain (0.1%) of TiNS-Gel ([TiNS] = 14 wt%) with keeping the temperature at 25 °C (**a**) and 90 °C (**b**). **c** Changes in the $G'$ and $G''$ values at a fixed frequency (1 rad s$^{-1}$) and a strain (0.1%) of TiNS-Gel ([TiNS] = 14 wt%) upon repeated thermal scanning between 25 and 90 °C.

electrostatic repulsion and (ii) the ultra-thin and extra-wide 2D shapes of its nanosheets, as described below.

TiNSs carry a dense array of negative charges (1.5 C m$^{-2}$) and they intrinsically show a large mutual electrostatic repulsion, which is essential in permitting the formation of TiNS-Gel$_{Repuls}$ (Fig. 1b–d). For the reversible transition between TiNS-Gel$_{Repuls}$ and TiNS-Gel$_{Attract}$, the electrostatic repulsion should be tunable (Fig. 1e–g)[26,27]. We found that the required tunability was provided by the thermoresponsive behavior of the TiNS surfaces, where tetrabutylammonium countercations (T$^+$) and protons (H$^+$) form ion pairs with surface anions (Ti–O$^-$), as shown in Fig. 6a. When an aqueous dispersion of TiNSs is heated from 25 to 90 °C, the countercations T$^+$, which are originally trapped by Ti–O$^-$ on TiNSs, are replaced with H$^+$ from bulk water. Because of this thermally induced T$^+$/H$^+$ replacement, the concentration of free T$^+$ ions increases from 0.25 mM (at 25 °C) to 0.55 mM (at 90 °C), as confirmed by ion conductometry (Fig. 6b, red). The thermally induced T$^+$/H$^+$ replacement also causes neutralization of the negative charges on Ti–O$^-$ on TiNSs, thereby lowering the absolute value of the TiNS surface potential from –60 mV (at 25 °C) to –36 mV (at 70 °C), as confirmed by measurements of the zeta potential (Fig. 6c, red). According to the DLVO theory[26,27], both of these heat-induced changes should contribute to an attenuation of the electrostatic repulsion between TiNSs, thereby explaining the mechanism of the transition from TiNS-

Gel$_{Repuls}$ into TiNS-Gel$_{Attract}$. Upon heating from 25 °C, the electrostatic repulsion between TiNSs is gradually attenuated and eventually, at ~55 °C, becomes weaker than the competing van der Waals attraction, so that TiNSs abruptly stack tightly to form an interconnected 3D network, resulting in a transition from TiNS-Gel$_{Repuls}$ into TiNS-Gel$_{Attract}$. Note that the thermoresponsive changes in the free-ion concentration (Fig. 6b) and surface-potential intensity (Fig. 6c) are reversible and show little hysteresis, which is consistent with the reversible and little-hysteresis nature of the transition between TiNS-Gel$_{Repuls}$ and TiNS-Gel$_{Attract}$, as confirmed by the SAXS (Fig. 3c) and rheological measurements (Fig. 4c).

To clarify the roles of another key property of TiNSs, their ultra-thin and extra-wide 2D shape, we constructed a thermal phase diagram of TiNSs dispersed in water ([TiNS] = 14 wt%) with systematic changes in the size of TiNSs (Fig. 6d). Aqueous dispersions of TiNSs of various sizes were prepared by sonicating an as-synthesized aqueous TiNS dispersion ([TiNS] = 0.4 wt%; original hydrodynamic size = 1.9 μm). The resulting TiNS dispersions were then concentrated to [TiNS] = 14 wt% by centrifugation as described above. When the size of TiNSs was sufficiently large (hydrodynamic size = 1.2–1.9 μm), the TiNS dispersion formed TiNS-Gel$_{Repuls}$ in the low-temperature region below 55 °C (Fig. 6d, #1, #2, #5, and #6) and TiNS-Gel$_{Attract}$ in the high-temperature region above 55 °C (Fig. 6d, #3, #4, #7, and #8). However, when the size of TiNSs was moderate (hydrodynamic size = 0.18–0.47 μm), the TiNS dispersion did not form TiNS-Gel$_{Repuls}$ in the low-temperature region but, instead, existed in a sol state (Fig. 6d, #9, #10, #13, and #14), probably because a reduction in the size of TiNSs enhanced their mobility, even though a strong electrostatic repulsion between TiNSs was present. In consistency with this, our previous report[28] on an aqueous dispersion of TiNSs with relatively small size (<~1 μm) showed a lower tendency of forming a lamellar architecture than that of this work. Meanwhile, in the high-temperature region, moderate-sized TiNSs (hydrodynamic size = 0.18–0.47 μm) were still capable of forming TiNS-Gel$_{Attract}$ (Fig. 6d, #11, #12, #15, and #16). When the size of TiNSs was further reduced (hydrodynamic size = 0.11 μm), the TiNS dispersion did not form a hydrogel in either the low-temperature region or the high-temperature region (Fig. 6d, #17–#20), probably because the size of TiNS was so small that their heat-induced aggregation did not result in an interconnected 3D network, but instead resulted in the possible formation of densely packed columnar stacks of TiNSs[40,41]. Overall, a large lateral size of TiNSs is essential for the formation of TiNS-Gel$_{Repuls}$ and TiNS-Gel$_{Attract}$, and the size of nanosheets required for the formation of TiNS-Gel$_{Repuls}$ (~1 μm) is much larger than that required for the formation of TiNS-Gel$_{Attract}$ (~0.18 μm).

## Discussion

We have developed a stimuli-responsive hydrogel (TiNS-Gel) consisting of inorganic constituents (14 wt% TiNSs and 86 wt% water). In response to temperature changes, TiNS-Gel reversibly adopts one of two hydrogel states (TiNS-Gel$_{Repuls}$ or TiNS-Gel$_{Attract}$) that have totally different internal structures: a lamellar architecture and an interconnected 3D network, respectively. The gel-to-gel transition proceeds within a narrow temperature range (55 ± 2 °C) and with excellent rapidity (within ~2 s) and perfect reversibility (little hysteresis and no deterioration upon repetition). The gel-to-gel transition was accompanied by marked changes in the optical properties (strong birefringence to no birefringence) and the mechanical properties ($G'$ = 26 kPa to $G'$ = 600 kPa) of the gel. Thus, the mechanical properties of TiNS-Gel changed considerably and rapidly as a result of

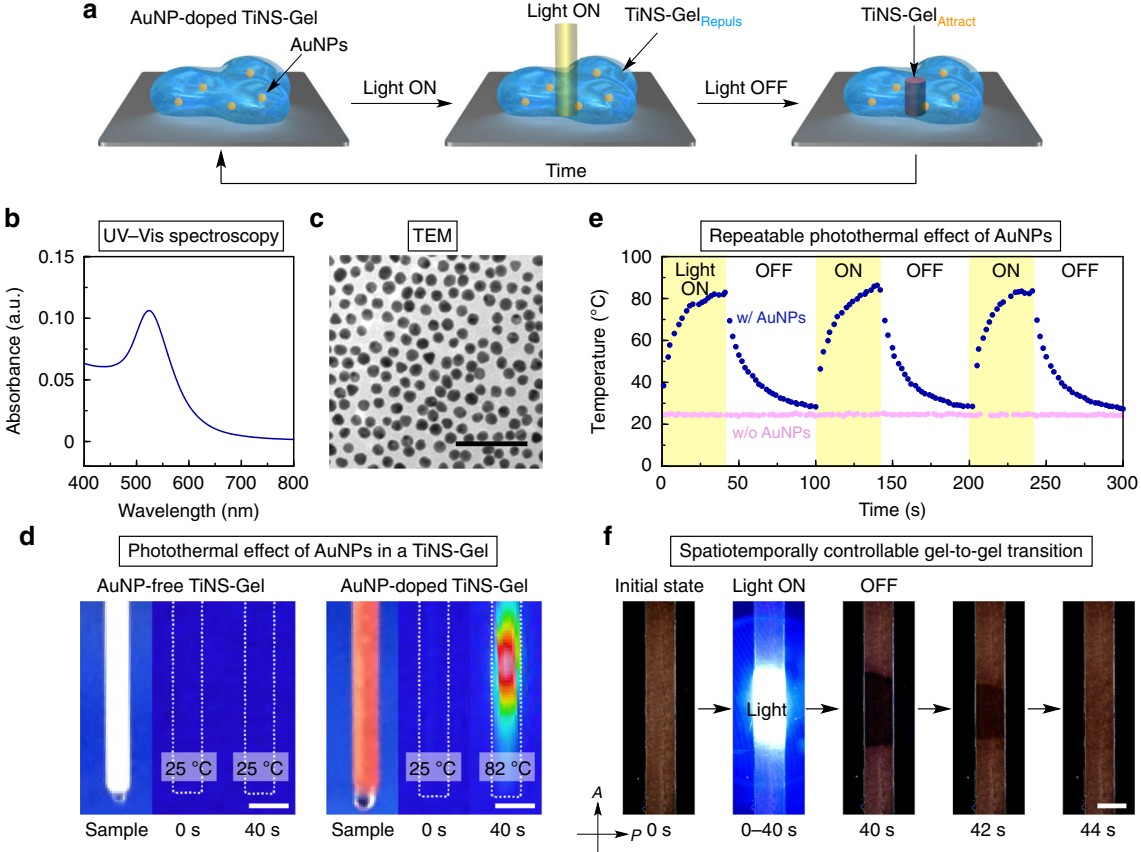

**Fig. 5 Spatiotemporally controlled transition between TiNS-Gel$_{Repuls}$ and TiNS-Gel$_{Attract}$ in response to photo stimuli. a** A schematic illustration for photoinduced gel-to-gel transition between TiNS-Gel$_{Repuls}$ and TiNS-Gel$_{Attract}$. **b, c** Characterization of gold nanoparticles (AuNPs; average diameter = 17 nm); UV-Vis absorption spectrum of an aqueous dispersion ([AuNP] = 0.005 wt%; **b**) and transmission electron microscopy image (TEM image; **c**). **d** Photoinduced temperature changes upon irradiation with 445-nm laser light (power density = 5.6 W cm$^{-2}$) of TiNS-Gel ([TiNS] = 14 wt%) without (left) and with (right) 0.05 wt% AuNPs. Pictures of the samples in a 1.5-mm diameter glass capillary (left) and their thermal-imaging camera images before (mid) and after (right) 40-s light irradiation. **e** Time-course temperature changes of TiNS-Gel ([TiNS] = 14 wt%) without (pink dots) and with (navy dots) 0.05 wt% AuNPs upon repeated turning on and off of the 445-nm laser light, which were monitored by a thermal-imaging camera. **f** Photoinduced changes in POM images under crossed Nicols of TiNS-Gel doped with AuNPs ([TiNS] = 14 wt%; [AuNP] = 0.05 wt%). When TiNS-Gel$_{Repuls}$ in a 0.2-mm-thick glass container was irradiated with the 445-nm laser light (irradiated region = 2 × 4 mm) for 40 s, the irradiated region was selectively transformed into TiNS-Gel$_{Attract}$ with no birefringence. Upon ceasing photoirradiation, the irradiated region returned to TiNS-Gel$_{Repuls}$ with a large birefringence within 4 s. This cycle could be repeated many times without deterioration (Supplementary Movie 1). Scale bars: 100 nm (**c**); 2 mm (**d**, **f**).

topological reconfiguration of its internal structure, in a manner reminiscent of sea cucumbers. By doping with gold nanoparticles as photothermal converters, the gel-to-gel transition could be induced by laser irradiation in a spatiotemporally controlled manner. To the best of our knowledge, no such smart material has previously been produced from inorganic materials exclusively.

The keys to this achievement are the characteristic profiles of TiNSs, which have an ultra-thin and extra-wide 2D shape and large and tunable electrostatic repulsion. Their shape endows TiNSs with the ability to form self-assembled 3D structures that contain abundant water efficiently, while their electrostatic properties permit the topology of these 3D structures to change in response to thermal stimuli. The 2D shape of TiNSs, which eliminates the possibility of entanglement, may be the reason for the excellent rapidity and reversibility of the thermal response of TiNS-Gel. These properties appear to be superior to those of conventional hydrogels based on 1D materials such as organic polymers or nanofibers[1–13]. Although poly(*N*-iso-propylacrylamide) and related organic polymers have long been the components of choice for smart soft materials[42,43], we have demonstrated that TiNS can serve as an alternative to such

organic polymers. This work will surely expand the scope of adaptive materials of the next generation, possibly even leading to the creation of 'inorganic life'.

## Methods

**Preparation of the hydrogel of TiNS (TiNS-Gel).** In a typical procedure, an aqueous dispersion (30 mL) of TiNS ([TiNS] = 0.4 wt%, [(C$_4$H$_9$)$_4$N$^+$OH$^-$] = ~90 mM) in a 50-mL centrifuge tube was centrifuged at 20,000 *g* for 1 h. From the resultant segregated mixture, the supernatant (~29 mL) was removed by pipetting, and the precipitate (0.86 mL) was collected as the hydrogel of TiNS in the repulsion-dominant state (TiNS-Gel$_{Repuls}$). Because the amount of TiNS included in the supernatant was negligibly small, the TiNS content in the resultant gel was calculated to 14 wt%. TiNS-Gel$_{Repuls}$ doped with AuNPs was prepared in the similar procedure, except for adding an aqueous dispersion of AuNPs ([AuNP] = 0.064 wt%, 0.67 mL) to the TiNS dispersion ([TiNS] = 0.4 wt%, 30 mL) before the centrifugation. Because the amount of AuNPs included in the supernatant was negligibly small, the AuNP content in the resultant gel was calculated to 0.05 wt%.

**Synthesis of gold nanoparticles (AuNPs).** An aqueous dispersion of gold nanoparticles (AuNPs) were synthesized according to the literature[33]. Typically, a solution of HAuCl$_4$·3H$_2$O in water (0.01 wt%, 50 mL) was heated to reflux. To the refluxing mixture, a solution of sodium citrate in water (1.0 wt%, 1.0 mL) was quickly injected. The color of the mixture turned to a brilliant red after 2 min, and its reflux for additional 3 min afforded an aqueous dispersion of AuNPs. Thus obtained AuNPs were monodisperse with an average diameter of 17 nm, as

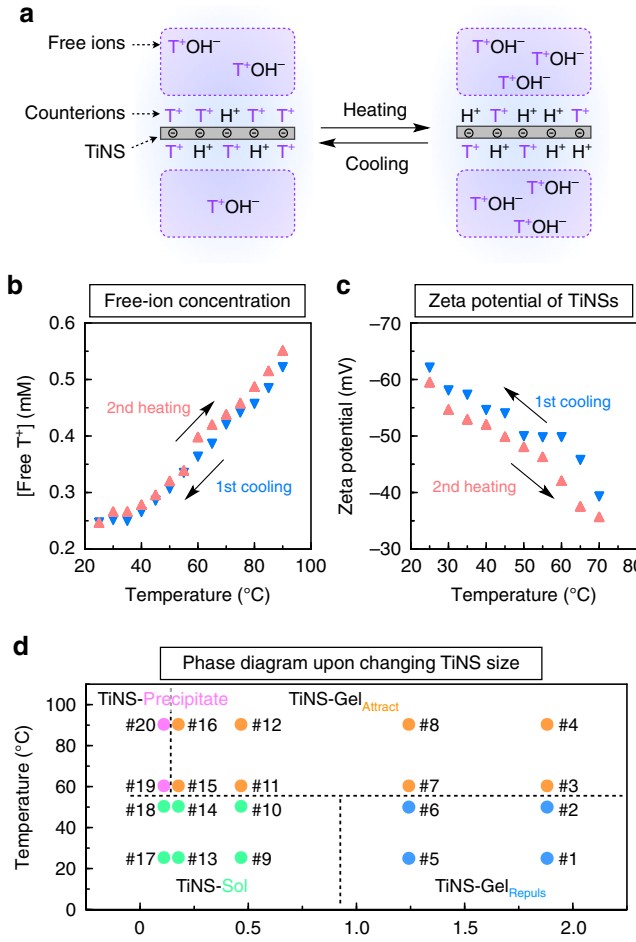

**Fig. 6 Mechanism of gel-to-gel transition and phase diagram of TiNS-Gel.**
**a** Schematic illustration for the mechanism of tunable electrostatic repulsion of TiNSs. TiNSs carry a lot of surface anions (Ti–O$^-$) that form ion pairs with tetrabutylammonium countercations (T$^+$) and protons (H$^+$). Upon heating, the countercations T$^+$, which are originally trapped by Ti–O$^-$ on TiNSs, are replaced with H$^+$ from bulk water. This thermally induced T$^+$/H$^+$ replacement increases the concentration of free T$^+$ ions and causes neutralization of the negative charges on Ti–O$^-$ on TiNSs. Consequently, upon heating, the electrostatic repulsion between TiNSs is attenuated. Upon cooling, the opposite changes occur, so that the electrostatic repulsion between TiNSs is recovered. **b, c** Changes in the concentration of free ions of T$^+$ (measured for a 0.1 wt% TiNS dispersion; **b**) and surface potential of TiNS (measured for a 0.01 wt% TiNS dispersion; **c**) upon thermal scanning. **d** Thermal phase diagram of TiNSs dispersed in water ([TiNS] = 14 wt%) with systematically changing the hydrodynamic size of TiNSs.

confirmed by TEM. Because AuNPs obtained in the above procedure could lose their colloidal dispersibility in the presence of TiNSs, AuNPs were coated with a polyethylene glycol ligand. Typically, thiol-appended polyethylene glycol mono-methyl ether (average molecular weight: 6 kDa; 120 mg) was added to an aqueous dispersion of AuNPs ([AuNP] = 0.005 wt%; 40 mL), and the mixture was mechanically shaken at 25 °C for 24 h. For removal of the unreacted reagent, the resultant mixture was subjected to repetitive centrifugation at 20,000 × g and subsequent re-dispersion with water to afford an aqueous dispersion of poly-ethylene glycol-coated AuNPs.

**Small-angle X-ray scattering (SAXS) analysis.** Two-dimensional SAXS mea-surements were carried out at BL45XU in the SPring-8 synchrotron radiation facility (Hyogo, Japan)[44] using Dectris model PILATUS3X 2 M detectors. The scattering vector $q$ ($q = 4\pi\sin\theta/\lambda$; $2\theta$ and $\lambda$ = scattering angle and wavelength of an incident X-ray beam [1.0 Å], respectively) and position of an incident X-ray beam on the detector were calibrated using several orders of layer reflections from silver

behenate ($d$ = 58.380 Å). The sample-to-detector distance was 2.5 m, where acquired scattering 2D images were integrated along the Debye–Scherrer ring, affording the corresponding one-dimensional profiles. A TiNS hydrogel sample held in a glass capillary (1.5 mm in diameter) was measured with gradually changing the temperature in a range of 25–90 °C (Supplementary Fig. 3).

**Rheological tests.** Rheological tests of TiNS gel were carried out by using an Anton Paar model MCR-301 rheometer. A hydrogel sample was set under a 25-mm diameter parallel plate with a sample gap of 1.0 mm. Storage and loss moduli ($G'$ and $G''$, respectively) were measured (i) on frequency sweep in a range of 0.1–10 rad s$^{-1}$ at a fixed strain of 0.1% with keeping the temperature at 25 °C (Fig. 4a and Supplementary Fig. 5b) and 90 °C (Fig. 4b and Supplementary Fig. 5c) and (ii) at a fixed frequency of 1.0 rad s$^{-1}$ and a fixed strain of 0.1% with changing the temperature in a range of a range of 25–90 °C at a rate of 1 °C min$^{-1}$ (Fig. 4c).

**Compression tests.** Compression tests of TiNS gel were carried out by using the same machine and set up as described for the rheological tests. Compressive stress was measured with gradually decreasing the sample gap from 1.0 to 0.99 mm at a rate of 0.02 mm min$^{-1}$ with keeping the temperature at 25 °C (Supplementary Fig. 4a) and 90 °C (Supplementary Fig. 4b).

## Data availability
The authors declare that the data supporting the findings of this study are available within the paper and its supplementary information files. All other information is available from the corresponding authors upon reasonable request.

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

## Acknowledgements

This work was financially supported by a JSPS Grant-in-Aid for Scientific Research (S) (18H05260) and a JSPS Grant-in-Aid for Scientific Research (B) (20H02791). We also acknowledge JST CREST Grant Number JPMJCR17N1, Japan. K.S. thanks JSPS for Grant-in-Aid for Research Activity Start-up (19K23642) and Young Scientists (20K15350) and Kurita Water and Environment Foundation (KWEF, Japan). K.S. also acknowledges RIKEN for the Special Postdoctoral Researcher Program. The small-angle X-ray scattering measurements were performed at BL45XU in SPring-8 with the approval of the RIKEN SPring-8 Center (proposal 20180067).

## Author contributions

K.S. conceived the project and designed the experiments. K.S. and N.I. performed all experiments. T.A. and Y.I. co-designed the experiments. Y.E. and T.S. prepared colloidally dispersed TiNSs. T.H. supported the small-angle X-ray scattering measurements at SPring-8. K.S. and Y.I. analyzed the data and wrote the manuscript with the input of all other authors. The manuscript reflects the contributions of all authors.

## Competing interests

The authors declare no competing interests.
