## [Peer Review File · Nature Communications]

REVIEWER COMMENTS

Reviewer #1 (Remarks to the Author):

The authors report stimuli-responsive hydrogel (TiNS-Gel), which can reversibly modulate the hydrogel states among the temperature changes. The author attributed the achievement to the properties of TiNSs with ultra-thin and extra-wide 2D shape, and large and tunable electrostatic repulsion.

- 1) The authors should provide more morphology characterization information of the as-prepared TiNSs, such as TEM and AFM.
- 2) The authors prepared different sizes of TiNSs by sonicating to clarify the effect of the size on the hydrogel states. However, there is no experimental data to verify whether the thickness of TiNSs affects the hydrogel states.
- 3) In order to produce a photoresponsive, the photothermal converter AuNPs was added to induce the photothermal-conversion ability of the composite hydrogel. However, I wonder whether the mutual interaction between the AuNPs and TiNSs affects the hydrogel states transition.
- 4) As an inorganic hydrogel, apart from the mechanical elasticity increased, are there any other significant changes and advantages among the gel-to-gel transition?

Reviewer #2 (Remarks to the Author):

This work describes interesting adaptive materials consisting of inorganic Ti-nanosheet structures. Lamellar aggregates of the sheets give rise to the formation of hydrogels that are able to reversibly switch mechanical stability while maintaining its gel state. In addition, this gel materials exhibit transient photothermal properties when doped with gold nanoparticles. Interestingly, the switching occurs rapidly without noticeable hysteresis, different from the organic hydrogels consisting of nanofiber networks. The thermo-responsive behavior of the gel materials is originated from delicate change in electrostatic properties of the Ti nanostructures, which is well characterized using appropriate techniques. I recommend this work published in Nat. Commun. after considering following questions.

- (1) The authors proposed that the gel-gel transition arises from the structural change from lamellar aggregates of several micrometer wide sheets at low T to interconnected nanofibers of 2.7 nm thickness at higher T. However, I could not see logical explanation how wide (several micrometer) sheets switch into very thin nanofibers of ~3nm thickness. Rolling, broken or stacking of the flat sheets? It looks like large structural change at the transition, as evidenced by the first order peak in DSC (Fig.3b). If the authors provide explanation with additional supporting data beyond X-ray on the structural transition from the flat 2D to interconnected nanofibers, the quality of the work will be highly improved. Along this line, I suggest the authors supply the TEM images in the text to make more convincing.
- (2) What is the colloidal stability at higher T compared with that at lower T?
- (3) Please check English. I also noticed that several Ref including vol no and page no are not correct. Please check.

Reviewer #3 (Remarks to the Author):

In this contribution Ishida, Aida and co-workers have demonstrated a unique example of all-inorganic dynamic and responsive hydrogel with remarkable spatio-temporal properties. Successful, responsive hydrogels often required tedious synthesis of organic components or a soft-hybrid approach by organic-inorganic co-assembly. And a purely inorganic-material based responsive hydrogels were beyond the imagination of supramolecular chemists. In this context,

the present contribution provides a breakthrough in the field of hydrogel research.

Authors have achieved responsive hydrogels by using anionic, thin nanosheets of titanate, and by exploiting its thermo-responsive reconfigurable microstructure. More importantly, by incorporating gold-nanoparticle as a photo thermal responsive components, authors could achieve a spatio-temporal responsiveness in these inorganic hydrogels. The experiments are performed in a well-organized manner and I am confident that these results open up a simple strategy to design responsive nanosheets.

Some of the points authors can address or comment is as follows.

1. Why the diffraction peak shifts with increasing temperature from 55°C to 90°C for TiNS-GelAttract in SAXS?
2. The authors mentioned that the volume of the gel remains unchanged during the transformation from TiNS-Gelrepuls to TiNS-GelAttract. how did they confirm it? did they perform Dilatometry?
3. How did the authors confirm the preserved properties of the hydrogel in both state after AuNP doping?
4. In figure 5 f) from the POM images the transformation from TiNS-Gelrepuls to TiNS-GelAttract under photo irradiation is taking 40s whereas upon ceasing photoirradiation, the irradiated region taking only 4s for the reverse transformation. Why this 10-fold decrease in time scale observed for the reverse transformation?
5. In the spatiotemporal experiment, the authors did not look into all the parameters extensively to control the dynamic structural change.
6. he authors are requested to check whether there is any effect of the power density of the irradiated light on the spatiotemporal change? Means, with increasing the intensity is it is taking more/less time to transform one gel state to another and in addition to that how much distance it is effecting from the center of the light source?
 - The authors are also suggested to check the role of % of AuNPs doping on spatiotemporal change?
7. Why the transition between TiNS-GelRepuls and TiNS-GelAttract shows hysteresis in SAXS and rheological measurement?

Perspective

1. Can authors comment on any other potential inorganic material which will perform in a similar manner?
2. Does magnetic alignment of these Nano sheets (as shown by the authors previously) will improve its performance?
3. I envisage lot of potential for these nanosheets, as a supramolecular scaffold for anchoring organic functional molecules to extract interesting responsive functions from the resultant solution processable soft-hybrids.

List of Additional Experiments

- (1) Supplementary Fig. 1: Transmission electron microscopy (TEM) imaging of TiNSs.
- (2) Supplementary Fig. 5: Rheological tests of AuNP-doped TiNS-Gel in the repulsive and attractive states.
- (3) Supplementary Fig. 6: Estimation of the transition speed of AuNP-doped TiNS-Gel from in the repulsive and attractive states upon photoirradiation.

Answers to Comments Raised by Reviewers #1:

The authors report stimuli-responsive hydrogel (TiNS-Gel), which can reversibly modulate the hydrogel states among the temperature changes. The author attributed the achievement to the properties of TiNSs with ultra-thin and extra-wide 2D shape, and large and tunable electrostatic repulsion.

(1) The authors should provide more morphology characterization information of the as-prepared TiNSs, such as TEM and AFM.

=> In response to this suggestion, we took TEM images of as-prepared TiNSs (Supplementary Fig. 1) and refer the data in the main text (page 5, line 5).

(2) The authors prepared different sizes of TiNSs by sonicating to clarify the effect of the size on the hydrogel states. However, there is no experimental data to verify whether the thickness of TiNSs affects the hydrogel states.

=> We appreciate this interesting question. However, TiNSs are unilamellar, and their thickness is uniformly determined to 0.75 nm by their atomic structure.

=> By using other nanosheets than TiNS, we can tune the thickness of constituent nanosheets of hydrogels, while it should also be noted that the properties of nanosheets other than thickness (width, hydrophilicity, surface charge, etc.) will also be changed. We would like to study on this issue in the future.

(3) In order to produce a photoresponsive, the photothermal converter AuNPs was added to induce the photothermal-conversion ability of the composite hydrogel. However, I wonder whether the mutual interaction between the AuNPs and TiNSs affects the hydrogel states transition.

=> To clarify this point, we newly measured the rheological properties of the AuNP-doped hydrogel, which were essentially identical to those of the AuNP-free hydrogel, irrespective of temperature (Supplementary Fig. 5). This result indicates that the interaction between AuNPs and TiNSs is negligible. To provide the above information, we added a description to the main text (page 9, line 17) and the experimental data to Supplementary Information (Supplementary Fig. 5)

(4) As an inorganic hydrogel, apart from the mechanical elasticity increased, are there any other significant changes and advantages among the gel-to-gel transition?

- => The gel-to-gel transition of TiNS-Gel is accompanied by significant changes in its birefringence, as shown in Figs. 2c, 2g and 3d–f.
- => The gel-to-gel transition of TiNS-Gel undergoes with excellent rapidity and reversibility, owing to the 2D shape of TiNSs that hardly entangle with each other, unlike the case of polymer chains in conventional thermoresponsive gels. For more detailed information, please also see Discussion section.

Answers to Comments Raised by Reviewers #2:

This work describes interesting adaptive materials consisting of inorganic Ti-nanosheet structures. Lamellar aggregates of the sheets give rise to the formation of hydrogels that are able to reversibly switch mechanical stability while maintaining its gel state. In addition, this gel materials exhibit transient photothermal properties when doped with gold nanoparticles. Interestingly, the switching occurs rapidly without noticeable hysteresis, different from the organic hydrogels consisting of nanofiber networks. The thermo-responsive behavior of the gel materials is originated from delicate change in electrostatic properties of the Ti nanostructures, which is well characterized using appropriate techniques. I recommend this work published in *Nat. Commun.* after considering following questions.

=> We appreciate these highly encouraging comments.

(1) The authors proposed that the gel-gel transition arises from the structural change from lamellar aggregates of several micrometer wide sheets at low T to interconnected nanofibers of 2.7 nm thickness at higher T . However, I could not see logical explanation how wide (several micrometer) sheets switch into very thin nanofibers of ~ 3 nm thickness. Rolling, broken or stacking of the flat sheets? It looks like large structural change at the transition, as evidenced by the first order peak in DSC (Fig. 3b). If the authors provide explanation with additional supporting data beyond X-ray on the structural transition from the flat 2D to interconnected nanofibers, the quality of the work will be highly improved. Along this line, I suggest the authors supply the TEM images in the text to make more convincing.

=> Thanks to this comment, we noticed that the schematic illustrations of the gel networks in the previous Fig. 1 were somehow misleading. In the illustrations at higher T (Fig. 1f, g), we did not intend to claim that TiNSs turned to nanofibers but that TiNSs stacked in a face-to-face manner to form an interconnected 3D network with maintaining their 2D shape. The supposed structure is supported by the SAXS (Fig. 2f) and SEM (Fig. 2h) measurements.

=> To avoid this possible confusion, we added a phrase “face-to-face stacking” to Fig. 1g.

(2) What is the colloidal stability at higher T compared with that at lower T ?

=> The colloidal stability of TiNSs decreases as T becomes higher, because of the heat-induced charge nebulization of TiNS surfaces. This tendency is clearly shown in the experimental data (Fig. 6 and Supplementary Fig. 2) and also rationally elucidated by the DLVO theory (Ref. 26 and 27). For detailed discussion, please see the main text on page 11, line 4–20.

(3) Please check English. I also noticed that several Ref including vol no and page no are not correct. Please check.

=> According to this kind comment, we carefully checked English and references throughout the manuscript.

Answers to Comments Raised by Reviewers #3:

In this contribution Ishida, Aida and co-workers have demonstrated a unique example of all-inorganic dynamic and responsive hydrogel with remarkable spatio-temporal properties. Successful, responsive hydrogels often required tedious synthesis of organic components or a soft-hybrid approach by organic-inorganic co-assembly. And a purely inorganic-material based responsive hydrogels were beyond the imagination of supramolecular chemists. In this context, the present contribution provides a breakthrough in the field of hydrogel research.

Authors have achieved responsive hydrogels by using anionic, thin nanosheets of titanate, and by exploiting its thermo-responsive reconfigurable microstructure. More importantly, by incorporating gold-nanoparticle as a photothermal responsive components, authors could achieve a spatio-temporal responsiveness in these inorganic hydrogels. The experiments are performed in a well-organized manner and I am confident that these results open up a simple strategy to design responsive nanosheets.

=> We appreciate these highly encouraging comments.

Some of the points authors can address or comment is as follows:

(1) Why the diffraction peak shifts with increasing temperature from 55°C to 90°C for TiNS-Gel_{Attract} in SAXS?

=> Judging from the observed TiNS distance (2.6 nm), together with the data reported in Ref. 28, not only the counteractions of TiNSs but also a small amount of water existed between TiNSs in TiNS-Gel_{Attract} at 55 °C. Therefore, the shift of the diffraction peak upon heating, indicating the slight decrease of TiNS distance, was probably because of the heat-induced dehydration of the counteractions and/or TiNS surfaces. To provide this information, we added a short description to the main text (page 7, line 19) with newly citing Ref. 39.

(2) The authors mentioned that the volume of the gel remains unchanged during the transformation from TiNS-Gel_{Repuls} to TiNS-Gel_{Attract}. How did they confirm it? Did they perform dilatometry?

=> During the gel-to-gel transition, we did not observe any exudation of water from TiNS-Gel, unlike the case of ordinary thermoresponsive hydrogels such as those composed of poly(-isopropylacrylamide), indicating that the volume of TiNS-Gel was maintained. To provide this information, we added a short description to the

main text (page 6, line 10).

- (3) How did the authors confirm the preserved properties of the hydrogel in both state after AuNP doping?

=> In response to this comment, we measured the rheological properties of AuNP-doped TiNS-Gel at 25 °C (repulsion-dominant state) and 90 °C (attraction-dominant state), which were almost identical to those of AuNP-free TiNS-Gel at both states (Supplementary Fig. 5). To provide this information, we added a short description to the main text (page 9, line 17) and the experimental data to Supplementary Information (Supplementary Fig. 5).

- (4) In Fig. 5f from the POM images the transformation from TiNS-Gel_{Repuls} to TiNS-Gel_{Attract} under photo irradiation is taking 40 s whereas upon ceasing photoirradiation, the irradiated region taking only 4 s for the reverse transformation. Why this 10-fold decrease in time scale observed for the reverse transformation?

=> In the experiment of Fig. 5f, we set the photoirradiation time as 40 seconds, simply for unifying the conditions of Fig. 5e and Fig. 5f. To evaluate how long photoirradiation is necessary for inducing the transition from TiNS-Gel_{Repuls} to TiNS-Gel_{Attract} in the experiment of Fig. 5f, we monitored the changes in birefringence of AuNP-doped TiNS-Gel upon photoirradiation and found that the transition was finished within 2 seconds (Supplementary Fig. 6). Thus, the time scales for the forward and reverse transitions were comparable to each other. To provide this information, we added a short description to the main text (page 10, line 11) and the experimental data to Supplementary Information (Supplementary Fig. 6).

- (5) In the spatiotemporal experiment, the authors did not look into all the parameters extensively to control the dynamic structural change.

The authors are requested to check whether there is any effect of the power density of the irradiated light on the spatiotemporal change? Means, with increasing the intensity is it is taking more/less time to transform one gel state to another and in addition to that how much distance it is effecting from the center of the light source?

=> In response to this comment, we monitored the changes in birefringence of AuNP-doped TiNS-Gel upon irradiation of lights with various power densities (0.56, 1.7, 2.8, and 5.6 W cm⁻¹) and found that the transition from TiNS-Gel_{Repuls} to TiNS-

Gel_{Attract} became faster as the light power density was increased (Supplementary Fig. 6b). To provide this information, we added a short description to the main text (page 10, line 11) and the experimental data to Supplementary Information (Supplementary Fig. 6b).

=> Because the light source we employ here is a laser, which is highly directional in general, it is expected that the transition speed is hardly affected by the distance between the sample and the light source.

The authors are also suggested to check the role of % of AuNPs doping on spatiotemporal change?

=> In response to this comment, we monitored the changes in birefringence of various types of AuNP-doped TiNS-Gel ([AuNP] = 0.05, 0.025, 0.01 and 0.005 wt%) upon photoirradiation and found that the transition from TiNS-Gel_{Repuls} to TiNS-Gel_{Attract} became faster as [AuNP] was increased (Supplementary Fig. 6c). To provide this information, we added a short description to the main text (page 10, line 11) and the experimental data to Supplementary Information (Supplementary Fig. 6c).

(6) Why the transition between TiNS-Gel_{Repuls} and TiNS-Gel_{Attract} shows hysteresis in SAXS and rheological measurement?

=> The transition between TiNS-Gel_{Repuls} and TiNS-Gel_{Attract} is a first-order phase transition. It is known that a first-order phase transition generally allows superheated/supercooled states and therefore exhibits a thermal hysteresis, unlike the case of a second-order phase transition.

Perspective:

(7) Can authors comment on any other potential inorganic material which will perform in a similar manner?

=> The synthesis of various metal oxide nanosheets other than TiNS have been reported. We believe that these nanosheets can also form hydrogels that undergo the similar gel-to-gel transition, using the design principle established in the present work. We will investigate this possibility in near future.

(8) Does magnetic alignment of these nanosheets (as shown by the authors previously) will improve its performance?

=> We appreciate this interesting idea. Magnetic alignment of the nanosheets should impart intriguing anisotropic properties to the resultant hydrogel.

(9) I envisage a lot of potential for these nanosheets, as a supramolecular scaffold for anchoring organic functional molecules to extract interesting responsive functions from the resultant solution processable soft-hybrids.

=> We appreciate this highly encouraging suggestion. As this reviewer points out, we also believe that our present work will expand design possibilities of responsive softmaterials.

REVIEWERS' COMMENTS

Reviewer #1 (Remarks to the Author):

The manuscript is acceptable now.

Reviewer #2 (Remarks to the Author):

I fully satisfy with the authors response and corrections on my questions, so that I recommend this work to be published as it is.

Reviewer #3 (Remarks to the Author):

I really appreciate the efforts made the by the authors to address all the comments in a satisfactory manner. I recommend the acceptance of this fine piece of work in the present form.

Reviewers #1 (Remarks to the Author):

The manuscript is acceptable now.

Reviewers #2 (Remarks to the Author):

I fully satisfy with the authors response and corrections on my questions, so that I recommend this work to be published as it is.

Reviewers #3 (Remarks to the Author):

I really appreciate the efforts made the by the authors to address all the comments in a satisfactory manner. I recommend the acceptance of this fine piece of work in the present form.

=> We are pleased that all the reviewers recommend our revised manuscript for publication in *Nature Communications*. We thank all the reviewers for taking the time to improve our paper.